# The impact of grandparent-grandchild interactions on the imagination of aging among Chinese youth groups: The chain mediating role of intergenerational relations and filial piety concept

Xiaojuan Hu[1]◉, Fen Xie[2]◉*

**1** Faculty of Education, GuangXi Normal University, GuiLin, Guangxi, China, **2** School of Journalism and Communication, GuangXi Normal University, GuiLin, Guangxi, China

◉ These authors contributed equally to this work.
* 1533657176@qq.com

## Abstract

Centered on respecting elders and intergenerational responsibility, Confucian filial piety culture serves as a core cultural norm in Chinese society. This study integrates this cultural framework with the context of China's rapid population aging to explore how grandparent-grandchild interactions within Chinese families shape young adults' aging imagination. Based on a survey of 778 Chinese youth aged 18–35, the findings show that both daily and ritualistic interactions positively predict young adults' aging imagination, and daily interactions have a stronger predictive effect. Intergenerational relations and filial piety not only serve as independent mediators but also constitute a chain mediating pathway, indicating that emotional bonds and cultural values synergistically foster young people's positive aging imagination. This study suggests that efforts to strengthen intergenerational interactions and integrate culturally rooted aging education are critical for promoting positive aging attitudes among youth.

## Introduction

Currently, China's aging process continues to accelerate. According to data from the Seventh National Population Census, the elderly population accounts for approximately 18.7% of China's total population, projected to reach one-third by the middle of this century. "Aging" refers to the aging structure of the entire population within a specific region.It concerns not only the elderly themselves but also involves the redistribution of resources among all members of society.This shift may trigger widespread social effects, impacting numerous areas including the labor market, industrial structure, social investment and consumption, and healthcare resources. From cultural and political perspectives, "aging" may affect people's mindsets, lifestyles, and alter the balance of interests and political dynamics among different age groups.

**Data availability statement:** The data that support the findings of this study are available at the following DOI link: https://doi.org/10.6084/m9.figshare.31819069.

**Funding:** This research was funded by Guangxi Philosophy and Social Science Foundation Project (23FSH028 to FX) and Guangxi Normal University Project (2024PY004 to FX).

**Competing interests:** The authors have declared that no competing interests exist.

Thus, constructing a positive aging society has become a key concern for Chinese policymakers and academic researchers. However, existing research in China primarily focused on the physical health, psychological state, or social integration of the elderly, with relatively less attention paid to the attitudes of the younger generation towards aging and the mechanisms behind these attitudes. Youth is a critical stage for the formation of values and cognitive models [1], and young people's perceptions and attitudes towards aging significantly affect the current and future societal acceptance of aging and the overall harmony of society. According to the Stereotype Embodiment Theory, stereotypes about the elderly persist throughout an individual's life. Negative stereotypes formed during youth can become part of self-perception, predicting psychological and physical conditions in later years. If negative stereotypes are internalized at a young age, they may lead to discrimination against the elderly, fear of one's own aging, and deteriorating health effects in old age [2]. Therefore, exploring factors influencing young people's perceptions of aging and identifying intervention pathways hold substantial theoretical and practical significance.

In the Chinese context, researchers have proposed the concept of "aging imagination", yet its specific connotations remain undefined. It has only been described as the perceptions of old age that non-elderly individuals gradually form through interactions with older adults [3]. Research on Self-Perceptions of Aging(SPA) provides a well-established theoretical foundation for understanding how individuals construct their own aging processes. Defined as individuals' cognitions, attitudes, expectations, and experiences regarding their aging trajectories, SPA explicitly encompasses both present-oriented evaluations and future-directed anticipatory elements [4,5]. However, existing SPA scholarship primarily focuses on the evaluative and experiential dimensions of aging, as well as their associations with health and developmental outcomes. In contrast, the psychological processes through which individuals construct future aging self-images via imagination and prospection have received scant conceptual attention. Notably, SPA research has not systematically elaborated at the theoretical level how aging-related expectations emerge, how they are symbolized, and how they ultimately integrate into a coherent vision of future elderly life.

Based on the SPA framework, we propose "aging imagination" as a distinct and theoretically grounded extension to capture the future-oriented, imaginative dimension of subjective aging. aging imagination is defined as individuals' psychological representations and narrative projections of their future elderly states, encompassing anticipated physical changes, social roles, life trajectories, and the meaning of later life. Unlike SPA, which primarily examines how individuals evaluate their own aging processes, aging imagination focuses on how individuals envision themselves growing old. Aging imagination serves as a preparatory self-cognitive framework that connects societal dominant perceptions of aging, individuals' personal views on aging, and behaviors when confronting aging.

Aging imagination is closely related to several well-established concepts in gerontology, yet it differs from them: attitudes toward aging, ageism, aging anxiety, and future time perspective. Attitudes toward aging generally refer to individuals' relatively stable evaluative orientations toward aging or the elderly stage, directed

either at aging as a life phase or at older adults as a group [6,7]. This concept primarily embodies a present-oriented value judgment or evaluative stance, often shaped by cultural norms and social discourses. While attitudes toward aging influence how individuals perceive the phenomenon of aging, they do not inherently involve how individuals imagine the old age they will experience in the future. Ageism denotes negative stereotypes and discriminatory behaviors targeting older adults as a social category [8,9]. It operates at the level of social norms and group attitudes, rather than focusing on individuals' internal perceptions of their future selves. Aging anxiety describes the emotional fear and distress individuals experience regarding their own aging process [10]. This concept emphasizes emotional reactions rather than structured expectations and preparatory behaviors. Future time perspective, by contrast, measures the extent to which individuals perceive their remaining lifespan as open-ended or limited, a perception that influences goal-setting and motivational orientations across the lifespan [11]. Although future time perspective serves as a critical temporal framework concept in the psychology of aging, it fails to elaborate on the specific connotations of aging imagination, an experience centered on self-identity, emotional acceptance, and preparatory behaviors.

In contrast to the aforementioned related constructs, aging imagination integrates individuals' cognitive content, emotional valence, and preparatory tendencies toward their own future aging. Defined as an integrative psychological structure, this concept bridges intergenerational experiences with young adults' subjective perceptions of their own later-life trajectories, thereby providing theoretical support for investigating how individuals perceive and anticipate the processes and outcomes of their own aging. This study distinguishes the concept of aging imagination from attitudes toward aging, ageism, aging anxiety, and future time perspective, with key differences summarized in Table 1.

Aging imagination is a subjective mental state that constructed in the interactions and closely related to social environment and cultural background [3]. In the Chinese context, addressing aging issues necessitates a return to the family as the primary unit. Familism is the foundation of traditional Chinese culture [12], and differs significantly from the highly "socialized" models seen in Western societies. In China, issues such as birth, aging, illness, and death are often regarded as family matters [13]. Intergenerational family interactions are also considered a key mechanism in shaping aging perceptions [14]. Existing research has shown that in intergenerational interactions, elderly individuals can improve the younger generation's understanding of aging and the elderly by transmitting life experiences, wisdom, and cultural skills [15]. However, several limitations persist in this field: First, most studies have primarily focused on the elderly population, with less attention paid to how young people acquire knowledge about aging through intergenerational interactions. Second, many studies focus on one-way intergenerational support(e.g., material or emotional support between generations), without exploring the differentiated impact of the forms of interactions (such as daily versus ritualized interactions). Third, the psychological mechanisms underlying intergenerational interactions remain unclear, particularly whether the quality of intergenerational relations(e.g., emotional closeness) and cultural values(e.g., filial piety) play a mediating role, which still requires further examination. In light of these gaps, this study is based on the Chinese family context and investigates the impact of grandparent-grandchild interactions on the younger generation's perceptions of aging, as well as the mechanisms through which these effects arise.

**Table 1. Comparison of Related Concepts.**

| Construct | Orientation | Temporal Focus | Core features |
|---|---|---|---|
| Aging Imagination (the core concept of this article) | Self | Future | The Psychological Structure of Future Self-Aging |
| Attitudes toward aging | Self/ General | Present | Evaluative judgments about aging |
| Ageism | Others | Present | Social stereotypes & prejudice |
| Aging anxiety, | Self | present | Emotional reactions to aging |
| Future time perspective | Self | Future | Perceived time horizon |

## Theoretical basis and research hypotheses

In the Chinese context, the family serves as the primary arena for the socialization of young individuals, providing them with the most intimate and authentic settings to witness and understand aging [16]. Both societal demographic aging and individual aging occur within the familial domain. The Chinese family is a rich tapestry of relations, where the ideal of "multiple generations under one roof" is highly valued [17]. Although increasing social mobility has impacted the physical manifestation of this ideal, the concept of "home" remains a source of comfort and emotional refuge for Chinese people [18]. Intergenerational relationships remain central to Chinese families, with multidimensional interactions profoundly shaping perceptions of aging and life. As life expectancy increases and population aging intensifies, Chinese families exhibit characteristics such as structural downsizing, intergenerational expansion, and diversification in family forms [19]. The communication between grandparents and grandchildren, alongside its cultural significance embedded within changing family structures, presents rich academic topics for exploration. Common modes of grandparent-grandchild communication include routine communication and ritualistic communication [20]. Routine communication refers to daily interactions characterized by high frequency, informality, and life-oriented features, such as watching television together, walking, or chatting, which involve caregiving and companionship [21]. Ritualistic communication occurs during traditional cultural occasions or life ceremonies and is marked by symbolic and normative characteristics, often bearing special cultural meanings [22]. For instance, during the Chinese New Year, activities such as pasting spring couplets, having reunion dinners, participating in ancestral rituals, and the exchange of red envelopes convey blood kinship, ethical values, and cultural identity in tangible forms. However, intergenerational communication within Chinese families is a dynamic, complex, and flexible process, whose influence on young people is marked by uncertainty, necessitating deeper analysis [23]. On one hand, interactions with the elderly allows younger generations to observe the aging process and acquire related knowledge potentially improving their negative stereotypes of aging [24]. Through intergenerational interactions, older adults impart life experiences, folk crafts, and cultural knowledge to the younger generation, aiding in the development of positive perspectives on aging [3] and life [25]. Some scholars suggest that more frequent intergenerational interactions ameliorate more positive views on aging among the youth [26]. On the other hand, intergenerational interactions could also induce fear of aging among younger generations. Some researchers argue that young people tend to view the elderly as outsiders, perceiving them as outdated and declining, with closed and lonely lifestyles, starkly contrasting with the vibrant characteristics of youth. In this context, instead of enhancing connection and integration, intergenerational interactions may exacerbate generational divides, age discrimination, and anxiety about self-aging [27]. Nussbaum J F and Coupland J, based on the Communication Accommodation Theory (CAT), examined the dialogic processes between older and younger generations. They suggested that communication barriers might arise due to unequal values and intergenerational relations, particularly when the elderly habitually converse from their perspective, with their manner and content not being understood or accepted by the younger, leading to ineffective intergenerational interactions [28].

Thus, we propose the following research hypothesis:

H1: Daily interactions between grandparents and grandchildren significantly influences the aging imagination of young people.

H2: Ritualistic interactions between grandparents and grandchildren significantly influences the aging imagination of young people.

## The mediating role of intergenerational relations between grandparents and grandchildren

Grandparent-grandchild contact serves as the initial experience of interactions between Chinese youth and the elderly, as well as the primary source of young people's perception of aging. Intergenerational relations often play a crucial role in influencing the effectiveness of intergenerational communication [16]. Unlike in Western societies, grandparental

caregiving and close intergenerational bonds are more common in China. Studies indicate that approximately 60% of Chinese families rely on grandparental caregiving, with the proportion being even higher in rural areas [29]. Grandparents perceive raising their grandchildren as a duty and responsibility, assuming the role of surrogate parents. From this perspective, the expansion of the grandparental role has become an undeniable reality [30]. This caregiving model fosters close emotional bonds between grandparents and grandchildren, which can persist into the grandchildren's adulthood [31]. Such intimate intergenerational relations are likely to promote positive perceptions, evaluations, and behavioral responses in social interactions [32]. When grandchildren maintain close emotional ties with their grandparents, they are more inclined to engage in interactions and explore their grandparents'inner worlds, thereby developing more positive perceptions of aging. As symbolic figures bridging the past, present, and future, grandparents offer young people invaluable opportunities to listen to their life experiences, understand their feelings, and learn about their concerns. This process helps the younger generation to comprehend philosophical themes such as memory, change, history, narrative, and the life course. It also allows them to appreciate the "life narratives" shared by the elderly, drawing wisdom and insights from these experiences. Empirical studies conducted in Taiwan suggest that emotional interactions between generations bridge the life stories of the elderly with the living environments of the youth. Older adults play a crucial role in transmitting experiences, knowledge, cultural values, and traditions. This intergenerational transmission enables young people to develop a more accurate understanding of aging [25].

Thus, we propose the following research hypothesis:

H3:Intergenerational relations mediate the association between daily grandparent-grandchild interactions and young peoples' aging imagination.

H4:Intergenerational relations mediate the association between ritualistic grandparent-grandchild interactions and young people's aging imagination.

## The mediating role of filial piety

Filial piety is the spiritual cornerstone of Chinese culture, encompassing the ethical consciousness of "respecting and serving elders" as well as the ancestral veneration and pursuit of continuity inherent in the patriarchal system. Under the discipline of filial ethics, individuals realize the value of their own lives by acknowledging the continuity of the family lineage [33]. Since the agrarian times, Chinese society has established a comprehensive set of behavioral norms related to filial piety, which accord the elderly a revered status [34]. Classical Chinese texts, including The Book of Songs (Shijing), Mozi, Laozi, Mencius, The Analects of Confucius, and The Book of Rites (Liji), contain extensive discussions on filial piety. Practices such as "respectfully serving elders" "inquiring about their well-being" "attending to parents with proper etiquette" and "entertaining parents with moral guidance" have been common interpersonal interactions between younger and older generations in Chinese families for thousands of years [35]. As a crucial element of Confucian ethics, filial piety has shaped the fundamental patterns of family communication in China. Although Chinese filial culture has faced challenges from globalization and social transformation, it remains highly resilient. Some scholars argue that traditional Chinese filial piety has evolved into a new form that aligns with modern civilized society [36]. In the current context, filial piety has not "declined" but has instead fostered a more stable cooperative model between generations [37].

As the most important traditional festival in China, the Spring Festival has long shaped the behavior of young people through the principles of filial piety. During this period, the ritual of "paying New Year's respects" is an essential tradition in every Chinese family, embodying the cultural value of "respecting the elderly". Elderly family members are the central figures and most honored participants in this ritual. During the process of paying respects, the younger generation formally offers greetings and blessings to their elders, reflecting the hierarchical order between generations deeply rooted in traditional Chinese culture [38]. In addition to the Spring Festival, Chinese society also places great emphasis on celebrating

the birthdays of elderly family members. These celebrations are not merely acknowledgments of the individual's longevity but also express reverence for the wisdom accumulated over a lifetime. Similar to the Spring Festival, birthday celebrations for the elderly are occasions for family reunions, during which younger family members demonstrate respect and gratitude through various gestures and rituals. These family ceremonies provide young people with emotional experiences and identity perceptions related to aging and the elderly. Such interactions foster more positive and stable interpersonal relations and group identity among family members [22], helping young people develop a more accepting attitude toward aging and death while reducing fears and anxieties associated with growing old [20].

Accordingly, the following hypotheses are proposed:

H5: Filial piety mediates the relationship between daily grandparent-grandchild interactions and young people's aging imagination.

H6: Filial piety mediates the relationship between ritualistic grandparent-grandchild interactions and young people's aging imagination.

### The chain mediating role of intergenerational grandparent-grandchild relations and filial piety

Intergenerational relations and filial piety are two key concepts in the study of Chinese families, and they are closely interconnected. Although in recent decades Chinese families have experienced trends of individualization similar to those in Western societies—leading to value and worldview differences between younger and older generations [12], a caregiving crisis [39], and challenges to traditional norms of filial piety—the notion of filial piety has not declined linearly. In fact, new characteristics have emerged. Emotional bonds between older and younger generations have grown stronger, and Chinese family relations have evolved into a form of familism characterized by individualistic values [40]. Some researchers categorize contemporary Chinese filial piety into reciprocal filial piety and authoritarian filial piety [41]. It has been observed that the better the intergenerational relationship, the more evident the reciprocal form of filial piety becomes. This positive development plays a crucial role in resolving intergenerational conflicts and other family issues [40].Conversely, strained intergenerational relations tend to weaken the younger generation's sense of filial obligation [33].Meanwhile, other studies have noted that although the influence of authoritarian filial piety is diminishing in Chinese families, it still retains a certain degree of relevance. This persistence can be attributed to the psychological inertia of the younger generation, who remain deeply influenced by traditional familism values [42].

Some researchers have analyzed the relationship between intergenerational relations and filial piety from the perspective of intergenerational cooperation. Amid the dual transformations of society and family, the rise of parental authority and the resurgence of filial piety have emerged as strategies for coping with uncertainties in social transitions, survival risks associated with individualization, and economic pressures. In this context, older adults have become crucial members of family reproduction, and the value of the elderly has been rediscovered [43]. Consequently, the ethical principle of filial piety continues to exhibit remarkable resilience and practical relevance [44]. In contemporary Chinese society, family intimacy serves as an essential resource for individuals in navigating risks and challenges [45]. This intergenerational relationship is not enforced by external forces, but sustained through psychological contracts rooted in familial ties and shared identity. This relationship subsequently shapes intergenerational negotiations and influences younger generations' perceptions of filial piety [45]. As Chinese families increasingly shift toward a nuclear family structure, intergenerational relationships exhibit more parallel characteristics, reflecting a significant change in the transmission and practice of filial piety in modern China [42].

Accordingly, the following hypotheses are proposed:

**H7:** Grandparent-grandchild intergenerational relations and filial piety play a chain mediating role between daily grandparent-grandchild interactions and young people's aging imagination.

**H8:** Grandparent-grandchild intergenerational relations and filial piety play a chain mediating role between ritualistic grandparent-grandchild interactions and young people's aging imagination.

Based on the above discussion, the following two research models are constructed (Figs 1 and 2):

## Method

### Ethics statement

This study has been approved by the Ethics Committee of Guangxi Normal University(No: 20240002). The research team clearly informed all participants about the study objectives, data usage, and confidentiality agreements prior to participation. Participants voluntarily participated after being fully informed of the study scope, purpose, and data management policy. Written informed consent was obtained from all participants, with explicit emphasis on their right to withdraw from the study at any time without consequences. All data were anonymized and securely stored, and were used solely for research purposes by the study team.

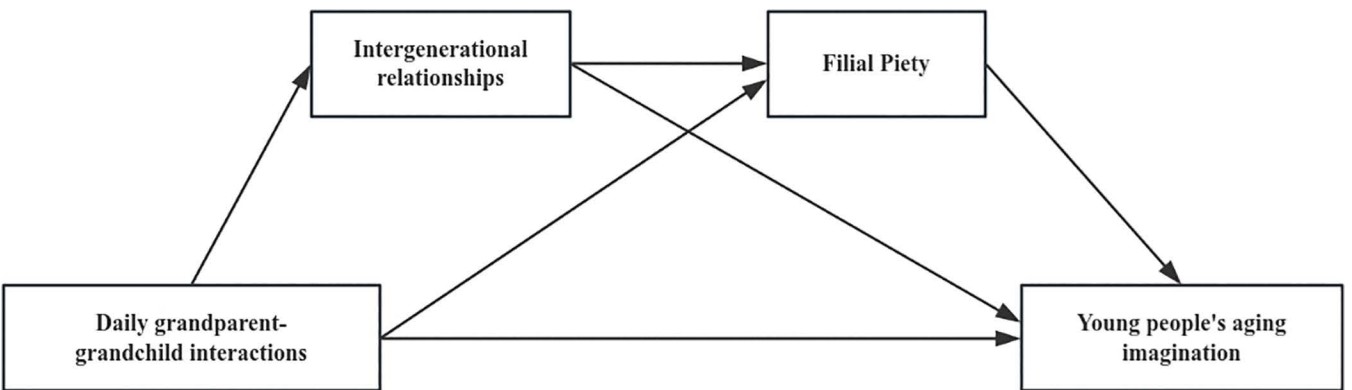

**Fig 1. The chain mediation model of daily interactions between grandparent- grandchildren and young adults' aging imagination(Model 1).**

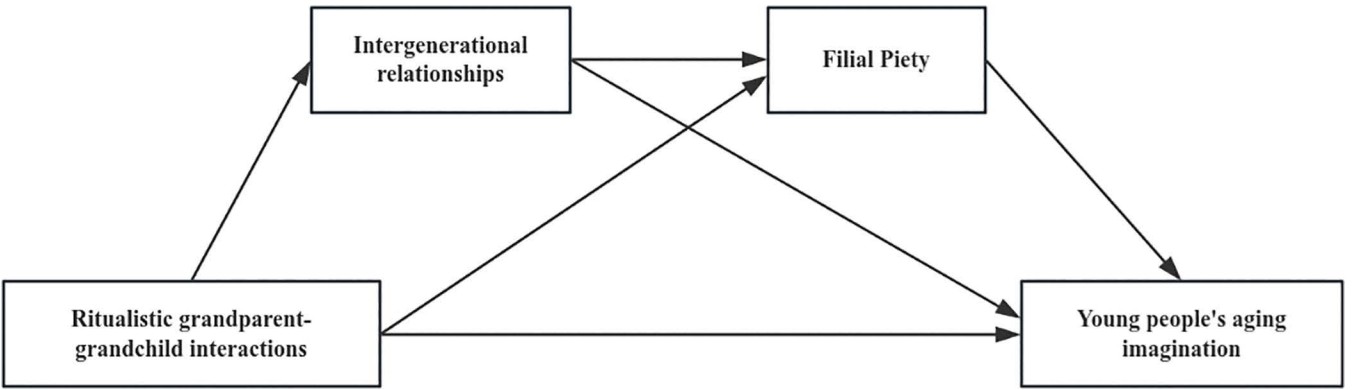

**Fig 2. The chain mediating model of ritualistic interactions between grandparent-grandchildren and young adults' aging imagination (Model 2).**

## Priori analyses

The required sample size (N) is calculated using the following formula (Kish, 1965).

$$N = \frac{Z^2 \times p \times (1-p)}{E^2}$$

In this formula, Z was set as the Z − value corresponding to the 95% confidence interval (Z = 1.96). The value of p was set at 0.5 to achieve the maximum sample size. E was defined as the width of the confidence interval, with a value of 5% adopted for the prediction. Consequently, n was approximately 385. We predicted a 70% response rate for completing the questionnaire, so a minimum sample size of 549 would be required.

## Recruitment and participants

The survey began to recruit participants on March 1, 2024 and ended on July 31.The participants were 800 young people from Guangxi Zhuang Autonomous Region, Guangdong Province and Shandong Province in China. Participants voluntarily joined the study after receiving detailed explanations of the research scope, purpose, and data handling policies. Written consent was obtained, emphasizing the right to withdraw at any stage without consequences.

When responses were incomplete and regular(i.e., of same answer or certain regularity), and the response time is less than 2 minutes, it reflected this questionnaire was invalid. After exclusion, a total of 778 participants were valid. Among the participants, 256 were male (32.8%) and 522 were female (67.2%). In terms of age distribution, 476 participants were aged 18–23, accounting for 61.18% of the sample, 191 were aged 24–29, representing 24.55%, and 111 were aged 30–35, comprising 14.27%.

## Measures

**Daily interactions between grandparents and grandchildren.** The scale measuring daily interactions between grandparents and grandchildren was adapted from Lin Qingshou's modified version. With a Cronbach's alpha coefficient of 0.849, this scale demonstrates a high degree of reliability [21]. Respondents were asked to answer four items on a 5-point Likert scale, which included: chatting with (paternal/maternal) grandparents, going shopping with (paternal/maternal) grandparents, accompanying (paternal/maternal) grandparents to the doctor, and having meals with (paternal/maternal) grandparents. In this study, the Cronbach's α for the scale was 0.87. The KMO value was 0.778. Bartlett's Test of Sphericity was significant, with an approximate chi-square value of 1919.863 (p < .001). These findings suggest that the data were suitable for factor analysis and that the measurement instrument demonstrated adequate reliability and construct validity.

**Ritualistic interactions between grandparents and grandchildren.** Respondents were required to answer two items on a 5-point Likert scale, specifically: having celebrated the birthday of (paternal/maternal) grandparents, and celebrating traditional family reunion festivals such as Chinese New Year with (paternal/maternal) grandparents.

**Intergenerational relations between grandparents and grandchildren.** The scale for measuring intergenerational relations was adapted from the version developed by Wang Honglei et al(Wang, Tan, & Bao, 2023). Respondents answered three items on a 5-point Likert scale, including: emotional closeness to (paternal/maternal) grandparents, having a good relationship with(paternal/maternal) grandparents, and willingness to listen when(paternal/maternal) grandparents have concerns or difficulties. The scale demonstrated strong internal consistency across each dimension, with a Cronbach's alpha value of 0.92(Wang, Tan, & Bao, 2023).In this study, the Cronbach's α for the scale was 0.89. The KMO value was 0.759. Bartlett's Test of Sphericity was significant, with an approximate chi-square value of 1750.095 (p < .001).

**Concept of filial piety.** Yeh distinguishes Confucian filial piety into reciprocal filial piety and authoritarian filial piety, but he does not conceptualize these two constructs as opposing or mutually exclusive value systems [46].In their subsequent research, the authors explicitly emphasized that these two filial piety motivations often coexist within the same individual and are co-embedded and intertwined in daily family practices, thus reflecting the integrative nature of filial piety in the Chinese cultural context [47]. When researching filial piety beliefs, scholars have noted that treating filial piety as an integrative construct better reflects its holistic cultural functions in Chinese family life [47](Ho, 1996; Bedford & Yeh, 2019). Therefore, we measured filial piety beliefs as a holistic construct using the scale developed by Taiwanese scholar Yeh. This instrument employs a 5-point Likert scale, with Cronbach's alpha coefficients ranging from 0.79 to 0.91, indicating good internal consistency. The items of the scale are as follows: paying attention to the health of the elderly, engaging in conversations with the elderly to understand their thoughts and feelings, being mindful of the daily life of the elderly, showing care and concern for the elderly, providing for the elderly to improve their comfort, feeling gratitude for the care received from the elderly, attending the funeral of the elderly regardless of the distance, helping the elderly when they are busy, yielding to the elderly when there is disagreement, handing over all earnings to the elderly before marriage, breaking promises to friends to comply with the elderly, giving up personal ambitions to fulfill the wishes of the elderly, promptly carrying out any task assigned by the elderly, avoiding marriage to someone disapproved of by the elderly, ensuring the continuation of the family line by having at least one son, and living with the elderly after marriage. In this study, the Cronbach's α for the scale was 0.886. The KMO value was 0.935. Bartlett's Test of Sphericity was significant, with an approximate chi-square value of 11890.020 ($p < .001$).

**Self-Imagined aging scale.** The Self-Imagined Aging Scale was adapted from a version developed by Taiwanese researcher Yuan Qiaozhen,with a cronbach's alpha value of 0.87 [48].The value indicated satisfactory to high internal consistency for the respective subscales. Respondents were asked to answer five items on a 5-point Likert scale, which included: being mentally prepared for the possibility of living a single life in old age, accepting the aging process and the changes in appearance, maintaining an optimistic view of aging, planning various strategies for the expected life in old age, and accumulating resources to achieve personal goals for aging. In this study, the Cronbach's α for the scale was 0.83.The KMO value was 0.758. Bartlett's Test of Sphericity was significant, with an approximate chi-square value of 1819.217 ($p < .001$).

## Data analysis

We conducted project analysis, reliability analysis, correlation analysis, and mediation effect test on the data using SPSS 22.0.

## Result

### Common method bias

In this study, Harman's single-factor test was employed to examine common method bias. Results indicated that all five factor eigenvalues exceeded 1, with the first factor accounting for only 29.31% of total variance, which is below the critical threshold of 40%. Secondly, given that the single-factor test may yield non-significant results under certain conditions, all measurement items corresponding to the study variables were loaded onto a single latent factor. A single-factor model was subsequently estimated using AMOS 26.0 to examine model fit. The results indicated a poor fit to the data, with a chi-square to degrees of freedom ratio ($\chi^2/df$) of 30.557, exceeding the recommended cutoff of 5. In addition, the goodness-of-fit index (GFI = 0.347), comparative fit index (CFI = 0.375), and Tucker–Lewis index (TLI = 0.329) were all below the acceptable threshold of 0.90, while the root mean square error of approximation (RMSEA = 0.195) exceeded the recommended upper limit of 0.08. Collectively, these findings demonstrate that the measurement items could not be adequately represented by a single factor, indicating that common method bias was not a serious concern in the present study.

## Correlation analysis

The results of the correlation analysis indicated that in Model 1(Table 2), significant correlations were observed among daily interactions between grandparents and grandchildren, intergenerational relations, filial piety, and young adults' aging imagination. Specifically, daily interactions was positively and significantly correlated with intergenerational relations ($r = 0.53$, $p < 0.01$), filial piety ($r = 0.33$, $p < 0.01$), and aging imagination ($r = 0.29$, $p < 0.01$). In Model 2(Table 3), significant correlations were also found among ritualistic communication between grandparents and grandchildren, intergenerational relations, filial piety, and aging imagination. Specifically, ritualistic interactions was positively and significantly correlated with intergenerational relations ($r = 0.45$, $p < 0.01$), filial piety ($r = 0.22$, $p < 0.01$), and aging imagination ($r = 0.25$, $p < 0.01$).

## The impact mechanism of daily interactions between grandparents and grandchildren on young adults' aging imagination

To examine the mechanism underlying young adults' aging imagination, this study utilized the Bootstrap test method proposed by Preacher, Rucker, and Hayes. Data were processed using the Process macro, with Model 6 selected as the primary analytical framework. In this model, the independent variable (daily interactions in Model 1 and ritualistic interactions in Model 2), mediators (intergenerational relations and filial piety), the dependent variable (young adults' aging imagination) and covariates (gender, age, education, family location, occupation, communication duration with grandparent(s)) were sequentially entered. The sample size was set at 5000, with a 95% confidence interval. The bias-corrected nonparametric percentile method was used for Bootstrap resampling, yielding the corresponding test results.The results indicated that daily interactions exerted a significantly positive impact on intergenerational relations ($\beta = 0.4645$, $p < 0.001$) as well as on the concept of filial piety ($\beta = 0.2007$, $p < 0.001$). Concurrently, daily interactions ($\beta = 0.1507$, $p < 0.001$), intergenerational relations ($\beta = 0.1640$, $p < 0.001$), and filial piety ($\beta = 0.1621$, $p < 0.001$) all contributed positively to young adults' aging imagination (Table 4, Fig 3).

In addition, ritualistic interactions demonstrated a positive effect on intergenerational relations ($\beta = 0.3852$, $p < 0.001$) and filial piety ($\beta = 0.0811$, $p < 0.05$). Furthermore, ritualistic interactions ($\beta = 0.1380$, $p < 0.01$), intergenerational relations

**Table 2. Model 1 Correlation analysis.**

| Variable | Mean±SD | 1 | 2 | 3 | 4 |
|---|---|---|---|---|---|
| 1.Daily grandparent-grandchild interactions | 3.25±0.96 | 1 | | | |
| 2.Intergenerational relations | 3.82±0.79 | 0.53** | 1 | | |
| 3.Filial Piety | 3.40±0.62 | 0.33** | 0.33** | 1 | |
| 4.Young adults' aging imagination | 3.75±0.77 | 0.29** | 0.30** | 0.27** | 1 |

Note.** $p < 0.01$. SD=standard deviation.

**Table 3. Model 2 Correlation analysis.**

| Variable | Mean±SD | 1 | 2 | 3 | 4 |
|---|---|---|---|---|---|
| 1.Ritualistic grandparent-grandchild interactions | 3.54±1.00 | 1 | | | |
| 2.Intergenerational relations | 3.82±0.79 | 0.45** | 1 | | |
| 3.Filial Piety | 3.40±0.62 | 0.22** | 0.33** | 1 | |
| 4.Young adults' aging imagination | 3.75±0.77 | 0.25** | 0.30** | 0.27** | 1 |

Note.** $p < 0.01$. SD=standard deviation.

**Table 4. Regression results of the chain mediating effects (Model 1).**

| Variables | Young adults' aging imagination | | Intergenerational relations | | Filial piety | | Young adults' aging imagination | |
|---|---|---|---|---|---|---|---|---|
| | β | t | β | t | β | t | β | t |
| Gender | −0.0212 | −0.5706 | −0.0541 | −1.6912 | −0.0185 | −0.5130 | −0.0076 | −0.2080 |
| Age | 0.0441 | 0.9458 | 0.0306 | 0.7621 | 0.0154 | 0.3412 | 0.0356 | 0.7829 |
| Education | −0.0061 | −0.1724 | 0.0262 | 0.8648 | −0.0096 | −0.2827 | −0.0097 | −0.2821 |
| Family location | 0.0543 | 1.5613 | −0.0136 | −0.4535 | 0.0078 | 0.2339 | 0.0557 | 1.6437 |
| Occupation | −0.0163 | −0.3554 | −0.0243 | −0.6172 | 0.0648 | 1.4637 | −0.0220 | −0.4915 |
| Communication duration with grandparent(s) | 0.0447 | 1.2207 | 0.2097 | 6.6659*** | 0.0559 | 1.5396 | −0.0058 | −0.1581 |
| Daily interactions | 0.2749 | 7.5189*** | 0.4645 | 14.7724*** | 0.2007 | 5.0209*** | 0.1507 | 3.6742*** |
| Intergenerational relations | | | | | 0.2063 | 5.1005*** | 0.1640 | 3.9496*** |
| Filial piety | | | | | | | 0.1621 | 4.4533*** |
| R | 0.2986 | | 0.5713 | | 0.3910 | | 0.3711 | |
| R² | 0.0892 | | 0.3264 | | 0.1529 | | 0.1377 | |
| F | 10.7684*** | | 53.3081*** | | 17.3524*** | | 13.6246*** | |

Note.*p < 0.1,** p < 0.01,***p < 0.001.

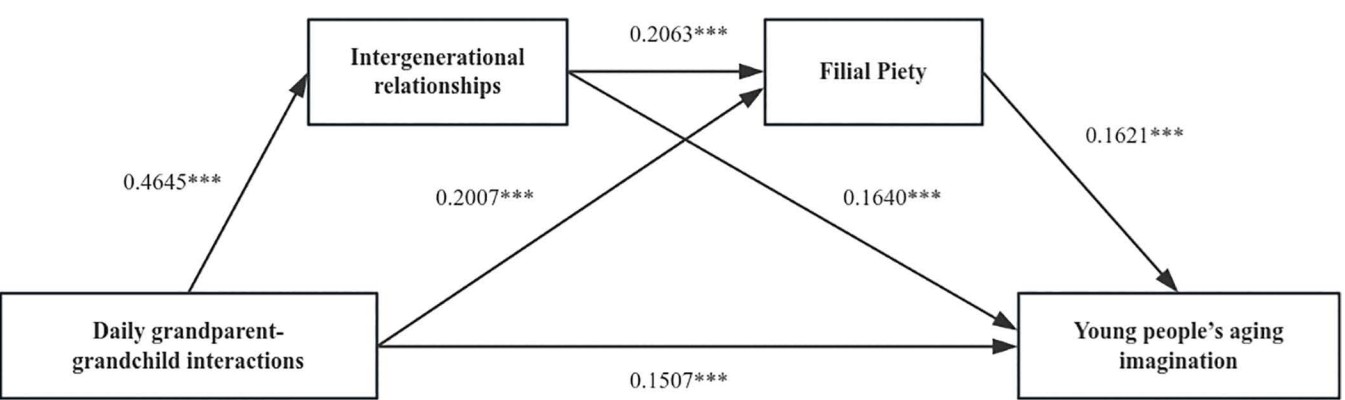

**Fig 3. The output result of the chained mediation for Model 1.**

(β = 0.1750, p < 0.001), and filial piety (β = 0.1761, p < 0.001) all exerted a positive influence on young adults' aging imagination (Table 5, Fig 4).

**Effects of grandparent-grandchild interactions on young adults' aging imagination.** To provide a more conservative estimate of the mediation effects, all results reported in the main text are based on models controlling for gender, age, education,family location, occupation,and communication duration with grandparents. For robustness checks, we also estimated the same serial mediation model without covariates, and the results were substantively similar. The corresponding results are reported in S1 Table and S2 Table in the Supporting information.

The results (see Tables 6 and 7) indicated that both daily interactions(β = 0.2749, p < 0.001) and ritualistic interactions(β = 0.2379, p < 0.001) between grandparents and grandchildren had significant effects on young adults' aging imagination, hypothesis1 and hypothesis2 were confirmed. The impact of these two pathways was positive, more frequent daily and ritualistic interactions was associated with a more positive aging imagination among young adults.

**Table 5. Regression results of the chain mediating effects (Model 2).**

| Variables | Young adults' aging imagination | | Intergenerational relations | | Filial piety | | Young adults' aging imagination | |
|---|---|---|---|---|---|---|---|---|
| | β | t | β | t | β | t | β | t |
| Gender | −0.0303 | −0.8054 | −0.0682 | −2.0439* | −0.0157 | −0.4304 | −0.0124 | −0.3390 |
| Age | 0.0393 | 0.8352 | 0.0231 | 0.5523 | 0.0130 | 0.2850 | 0.0319 | 0.7008 |
| Education | −0.0129 | −0.3622 | 0.0152 | 0.4836 | −0.0134 | −0.3878 | −0.0139 | −0.4047 |
| Family location | 0.0669 | 1.9036 | 0.0064 | 0.2054 | 0.0108 | 0.3157 | 0.0636 | 1.8714 |
| Occupation | −0.0123 | −0.2666 | −0.0179 | −0.4367 | 0.0677 | 1.5091 | −0.0203 | −0.4527 |
| Communication duration with grandparent(s) | 0.0753 | 2.0926* | 0.2659 | 8.3279*** | 0.0795 | 2.1820* | 0.0022 | 0.0606 |
| Ritualistic interactions | 0.2379 | 6.5696*** | 0.3852 | 11.9912*** | 0.0811 | 2.1189* | 0.1380 | 3.6084** |
| Intergenerational relations | | | | | 0.2686 | 6.8154*** | 0.1750 | 4.3276*** |
| Filial piety | | | | | | | 0.1761 | 4.8993*** |
| R | 0.2724 | | 0.5211 | | 0.3609 | | 0.3703 | |
| R² | 0.0742 | | 0.2716 | | 0.1302 | | 0.1372 | |
| F | 8.8143*** | | 41.0076*** | | 14.3920*** | | 13.5639*** | |

Note.*p<0.1,** p<0.01,***p<0.001.

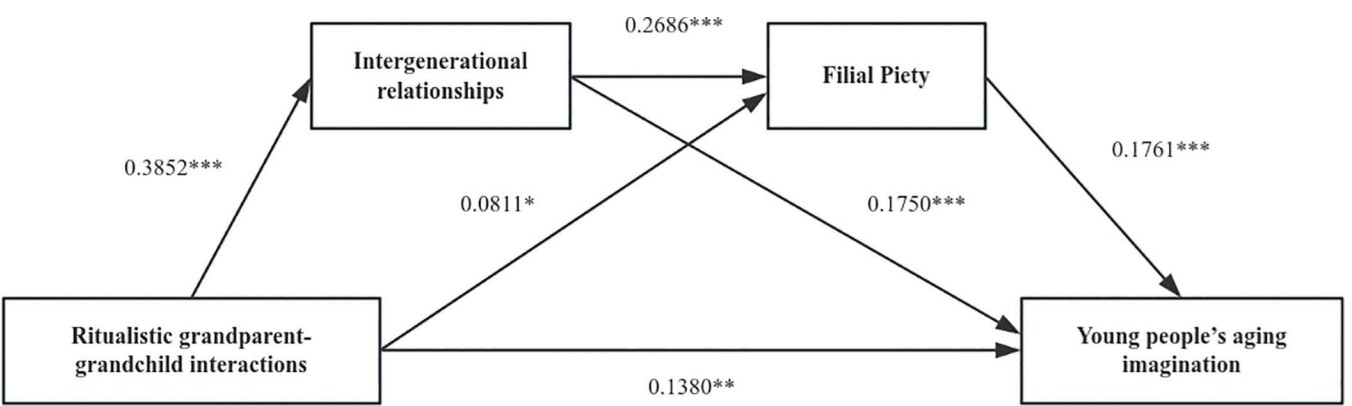

**Fig 4. The output result of the chained mediation for Model 2.**

**Table 6. The chain mediation of interpersonal relations and filial piety among young people(Model 1).**

| Model | Intermediary paths for variables | Standardized effect size | BootSE | 95%CI BootLLCI | BootULCI |
|---|---|---|---|---|---|
| Model 1 | D.G.G.I→Y.P.I.A(Total Effect) | 0.2749 | 0.0366 | 0.2031 | 0.3467 |
| | D.G.G.I→Y.P.I.A(Total Direct effect) | 0.1507 | 0.0410 | 0.0702 | 0.2312 |
| | Total Indirect Effect | 0.1242 | 0.0283 | 0.0700 | 0.1827 |
| | D.G.G.I→I.R→Y.P.I.A | 0.0762 | 0.0234 | 0.0306 | 0.1236 |
| | D.G.G.I→F.P→Y.P.I.A | 0.0325 | 0.0123 | 0.0120 | 0.0612 |
| | D.G.G.I→I.R→F.P→Y.P.I.A | 0.0155 | 0.0061 | 0.0060 | 0.0295 |

Note. D.G.G.I,Daily grandparent-grandchild interactions; Y.P.I.A,Young people's imagination of aging; I.R,Intergenerational relations; F.P,filial piety.

**Table 7. The chain mediation of interpersonal relations and filial piety among young people (Model 2).**

| Model | Intermediary paths for variables | Standardized effect size | BootSE | 95%CI | |
|---|---|---|---|---|---|
| | | | | BootLLCI | BootULCI |
| Model 2 | R.G.G.I→Y.P.I.A (Total Effect) | 0.2379 | 0.0362 | 0.1668 | 0.3090 |
| | R.G.G.I→Y.P.I.A(Total Direct effect) | 0.1380 | 0.0382 | 0.0629 | 0.2131 |
| | Total Indirect Effect | 0.0999 | 0.0225 | 0.0593 | 0.1470 |
| | R.G.G.I→I.R→Y.P.I.A | 0.0674 | 0.0192 | 0.0324 | 0.1079 |
| | R.G.G.I→F.P→Y.P.I.A | 0.0143 | 0.0083 | 0.0005 | 0.0328 |
| | R.G.G.I→I.R→F.P→Y.P.I.A | 0.0182 | 0.0062 | 0.0080 | 0.0319 |

Note. R.G.G.I,Ritualistic grandparent-grandchild interactions; Y.P.I.A,Young people's imagination of aging; I.R,Intergenerational relations; F.P,filial piety.

**Mediating role of intergenerational relations.** The results(see Tables 6 and 7) further demonstrated the mediating effect of intergenerational relations. In Model 1, the indirect effect of daily interactions on aging imagination through intergenerational relations was 0.0762, with a 95% Bootstrap confidence interval of [0.0306, 0.1236], which did not include zero. This indicates that intergenerational relations significantly mediated the effect of daily interactions on young adults' aging imagination. Specifically, more frequent daily interactions enhanced intergenerational relations, thereby fostering a more positive aging imagination. Thus, hypothesis3 was supported.

In Model 2, the indirect effect of ritualistic interactions on aging imagination through intergenerational relations was 0.0674, with a 95% Bootstrap confidence interval of [0.0324, 0.1079], which also did not include zero. This suggests that intergenerational relations significantly mediated the relationship between ritualistic interactions and young adults' aging imagination. In particular, ritualistic interactions improved intergenerational relations, which in turn positively influenced aging imagination. Therefore, hypothesis4 was supported.

**Mediating role of filial piety.** The findings (see Tables 6 and 7) revealed that in Model 1, the indirect effect of daily grandparent-grandchild interactions on young adults' aging imagination through filial piety was 0.0325, with a Bootstrap confidence interval of [0.0120, 0.0612], excluding zero. This indicates a significant mediating effect of filial piety in the relationship between daily interactions and aging imagination. Specifically, frequent daily interactions strengthened filial piety among young adults, which in turn fostered a more positive attitude toward aging. Therefore, hypothesis5 was supported.

In Model 2, the indirect effect of ritualistic interactions on aging imagination through filial piety was 0.0143, with a Bootstrap confidence interval of [0.0005,0.0328], also excluding zero. This suggests that ritualistic interactions positively influenced young adults' aging imagination through the enhancement of filial piety. Accordingly, hypothesis6 was supported.

**Serial mediation of intergenerational relations and filial piety.** The results (see Tables 6 and 7) demonstrated a significant serial mediation effect of intergenerational relations and filial piety in the relationship between grandparent-grandchild interactions and young adults' aging imagination.

In Model 1, the serial mediation effect for daily interactions was 0.0155, with a Bootstrap confidence interval of [0.0060,0.0295], excluding zero. This indicates that daily interactions improved the quality of intergenerational relations, which subsequently enhanced young adults' filial piety, leading to more positive aging imagination. Therefore, hypothesis7 was supported.

In Model 2, the serial mediation effect for ritualistic interactions was 0.0182, with a Bootstrap confidence interval of [0.0080,0.0319], excluding zero. This finding suggests that ritualistic interactions strengthened intergenerational relations and, consequently, heightened filial piety, fostering more positive aging imagination. Thus, hypothesis8 was supported.

## Discussion

This study, grounded in the context of Chinese families, explored the mechanisms through which daily and ritualistic grandparent-grandchild interactions influences young adults' aging imagination. The findings underscore the significant role of intergenerational contact in shaping young adults' attitudes toward aging. It was found that both daily and ritualistic interactions positively predicted aging imagination directly and indirectly through intergenerational relations and filial piety. Additionally, the serial mediation effects of intergenerational relations and filial piety were confirmed, offering deeper insights into the mechanisms of grandparent-grandchild interactions. These findings contribute to ongoing dialogues with existing theories, such as stereotype internalization theory, and provide valuable implications for the fields of communication studies and gerontology.

### Dual pathways of grandparent-grandchild interaction shaping young adults' aging imagination: daily interactions vs ritualistic interactions

The positive impacts of daily interaction on young adults' aging imagination. Social Learning Theory posits that individuals form cognitions and attitudes through observing and imitating others' behaviors, as well as receiving environmental feedback [49]. Daily interactions provides young adults with high-frequency, authentic contexts for observing aging: grandparents' aging coping strategies and positive life attitudes, demonstrated through interactions such as daily conversations, shared household chores, and medical accompaniment, serve as direct behavioral models of aging that young adults can emulate. This contextualized learning is immediate and sustained, and the portrayal of aging in daily interactions aligns more closely with real life [50]. Harwood's(2000) research on intergenerational communication also revealed that frequent daily contact reduces young adults' sense of unfamiliarity and fear toward aging. In contrast, the unique cultural and symbolic meanings embedded in ritualistic communication may require more time and higher-frequency activities to exert a more substantial impact. [15]. Additionally, Intimacy Theory suggests that high-frequency, informal interactions are more conducive to strengthening emotional trust and psychological bonding [51]..Daily interactions is often accompanied by more reciprocal role behaviors (e.g., older generations providing care to young adults, or young adults accompanying older adults to medical appointments). Such bidirectional support reinforces intergenerational functional solidarity [52], allowing younger generations to more intuitively perceive that aging is not a one-way dependent process but rather a natural and valuable stage in the life course [53]. With the penetration of digital media into Chinese families, information and communication technologies (ICTs), such as social media and smart monitoring systems, are reshaping family interactions and daily life by constructing increasingly sophisticated social scenarios, a process infused with the interplay of emotions and power [54]. From urban to rural contexts, and from quantitative panel data analyses to qualitative in-depth interviews, studies have repeatedly confirmed that mediated interactions with Chinese characteristics exert a significant impact on intergenerational relationships. Such interactions have emerged as new modes of intergenerational engagement in everyday life and serve as a vital supplement to family communication [55–57].

Ritual interaction also exerts an undeniable shaping effect on young adults' aging imagination. In Chinese contexts, rituals such as the elderly's birthdays and traditional festivals often extend beyond the nuclear family, evolving into collective events that incorporate all participating relatives into the broader "family" category [58]. Within these extended family ritual settings, older adults serve as pivotal emotional and cultural bonds [59]. The theory of family solidarity posits that family members' identification with family norms and the intimacy of their emotional ties directly influence the occurrence and intensity of mutual support behaviors. As key carriers of family norms, Chinese older adults naturally convey cultural values such as "respect for hierarchy between elders and juniors" and "mutual assistance" during rituals [60]. Leveraging their position as the emotional core, they maintain the stability of family emotional solidarity while strengthening connections among family members through intangible support, such as life wisdom and advice. Studies have shown that the frequency of contact among relatives tends to decrease significantly after the passing of older adults [59], which indirectly

confirms the central role of older adults in sustaining family networks. In rituals like birthdays and festivals, young adults enter a culturally constructed identity space by participating in gift preparation, ritual practice, family storytelling, and other links. Grandparents not merely as individual relatives, but as symbolic carriers of family history, cultural inheritance, and life continuity [61]. Long-term engagement and observation allow young adults to witness the ongoing contributions of elders within the family, which in turn activates their concept of filial piety and helps them recognize that old age is not merely a stage of one-way dependence and demand, but a period still endowed with the potential to create emotional and cultural value [62].

Notably, the relative effects of these two communication forms may be moderated by a range of family contextual and structural variables, an area warranting further in-depth exploration in future research. First, residential arrangements and geographic proximity: Co-residence or living in close proximity is likely to significantly enhance the accessibility and emotional intensity of daily interactions [63], thereby influencing interactions effects. In contrast, long-distance separation may compel ritualistic interactions (e.g., returning to one's hometown during the Spring Festival) to fulfill a stronger emotional compensation function [64].Second, migration status and family mobility: For young migrants or family members living away from their hometowns, digital media serves as a crucial supplement to face-to-face communication, fostering the emergence of new digital rituals and thereby shaping interactions outcomes [65].

In summary, daily interactions holds an advantage in emotional learning and cognitive restructuring through high-frequency, intimate interactions, while ritualistic interactions plays an irreplaceable role in activating cultural values and reinforcing norms. Together, they constitute dual pathways shaping young adults' perceptions of aging, with their effectiveness embedded within specific family and social contexts.

## Mediating role of intergenerational relations: psychological mechanisms of intergenerational interactions

The findings indicate that the grandparent-grandchild intergenerational relationship plays a significant mediating role in shaping young adults' perceptions of aging, both in their daily interactions and Ritual interaction with grandparents. This reveals the psychological pathway through which intergenerational interactions influence young people's cognition of aging: communication behaviors first affect the quality of emotional bonds between generations (i.e., intergenerational relationships), which in turn shapes young adults' future perceptions of their own aging via these emotional ties. This discovery advances research beyond a sole focus on interaction frequency or forms to an exploration of the relational dynamics embedded in such interactions. Furthermore, the prevalent phenomenon of "grandparent-grandchild bond" in Chinese society provides a unique and crucial cultural context for understanding this psychological mechanism.

Within Chinese families, "grandparent-grandchild bond" stands as a typical phenomenon, often fostering a natural intergenerational bond between grandparents and grandchildren characterized by high emotional involvement [29]. Grandparents play a pivotal role in the upbringing of their grandchildren [30], and the profound emotional connections established during this early stage lay the groundwork for grandparent-grandchild communication in adulthood [31]. The present study confirms that both daily and ritualized communications further function on this basis. This positive relational state rooted in "grandparent-grandchild bond"constitutes an especially safe and intimate psychological context. When observing their grandparents' aging process within this context, young adults are more likely to experience emotional resonance and identity alignment, rather than alienation and fear. The aging of grandparents becomes closely intertwined with the warm image of "loving grandparents", thereby effectively de-alienating and de-stigmatizing societal stereotypes about old age [66]. Emotional solidarity represents a key dimension of intergenerational solidarity [67], and"grandparent-grandchild bond" serves as a quintessential manifestation of such emotional solidarity in the Chinese context, facilitating positive intergenerational perceptions.

Further integrating Socioemotional Selectivity Theory [11], interactions within the "grandparent-grandchild bond"relationship tend to be more purely oriented toward emotional fulfillment. In this type of relationship, young adults are more likely to perceive their grandparents as caregivers and stable anchors of family emotion. Taiwanese scholars noted that

emotionally intimate intergenerational relationships form the foundation for the effective transmission of life stories [25]. Against the backdrop of Chinese "grandparent-grandchild bond," the life stories and wisdom shared by grandparents are not merely general narratives but special legacies infused with the emotional warmth of the family [62]. What young adults gain from these narratives is not only knowledge about aging but also a sense of life continuity rooted in being loved and accepted. This enables them to envision their own future old age as a phase where they can continue to give love, pass on wisdom, and maintain family emotional bonds, thereby constructing a more emotionally connected and value-laden "perception of aging" [68].

In summary, the mediating role of intergenerational relationships is reinforced and concretized within the cultural reality of "grandparent-grandchild bond" in China. It indicates that young Chinese adults' positive perceptions of aging are rooted in a unique family emotional structure grounded in intergenerational ties.

## Mediating role of filial piety: pathways of cultural values

The mediating effect of filial piety highlights the significant influence of cultural values on intergenerational interactions. This study supports the theory of "bilateral socialization" in filial piety ethic, where young adults not only internalize traditional filial norms but also actively practice them. Filial piety emphasizes respect and dignity for the elderly, viewing them as valued individuals rather than a vulnerable group [69]. In Chinese culture, filial piety is a cornerstone of intergenerational solidarity [70]. Young adults who internalize filial piety are more likely to engage in meaningful interactions with the elderly, enhancing their understanding and respect for aging as a natural and valued process. Compared to daily interactions, the effect of filial piety was smaller in ritualistic interactions, possibly due to the formalization of rituals in modern society [15]. At the same time, some scholars argue that filial piety in China exerts a certain degree of pressure on young people. Overemphasizing the elderly's authoritative status and ritualized intergenerational communication may also impose moral burdens on young individuals, under such pressure, the positive impact of ritual interactions is undermined [71].

## Serial mediation: synergistic effects of intergenerational relations and cultural values

The findings demonstrate that intergenerational relationships and filial piety beliefs form a significant chain mediating pathway between grandparent-grandchild interactions (both daily and ritualized) and young adults' aging imagination. This not only verifies the respective mediating roles of the two but also reveals the dynamic synergistic mechanism between them: intimate intergenerational relationships provide an emotional foundation for the internalization of filial piety cultural values, while internalized filial piety values in turn influence intergenerational interactions, thereby, shape young adults' positive aging imagination. This discovery integrates emotional bonds and cultural norms from two parallel explanatory variables into a sequential action model, profoundly uncovering the sequential pathway of intergenerational influence within Chinese families.

First, high-quality grandparent-grandchild relationships facilitate the acceptance of filial piety beliefs. According to Attachment Theory, secure emotional bonds are more conducive to individuals' acceptance and internalization of external norms [72]. Positive intergenerational relationships transform the filial piety demands perceived by young adults from authoritative discipline into natural responses and emotional expressions of family affection. This aligns with Yeh and Bedford's (2003) definition of "reciprocal filial piety", which posits that the motivation for reciprocal filial piety stems from emotional gratitude and the desire to repay the nurturing kindness of parents/grandparents [46]. Thus, positive intergenerational relationships convert filial piety into a relational practice closely tied to personal emotions. Subsequently, the emotionally internalized filial piety beliefs further enhance the positive construction of aging imagination. When filial piety beliefs are accepted emotionally, they elevate from a set of behavioral guidelines to a cognitive and meaning-making framework. Young adults reinterpret their grandparents' aging process through the lens of "respecting and honoring the elderly": grandparents' physical changes are regarded as a natural life course, and their life wisdom is recognized as a transmissible cultural resource [68].

This chain pathway also vividly reflects the interaction between emotional solidarity and normative solidarity in the Intergenerational Solidarity Theory [67]. The present study finds that emotional closeness (intergenerational relationships) reinforces identification with family norms (filial piety beliefs), namely, normative solidarity, and such norm practice based on identification, in turn, consolidates and deepens emotional bonds. Against the backdrop of Chinese "grandparent-grandchild bond" and familism culture, this cyclic process is confirmed by the study: intimate grandparent-grandchild relationships create conditions for the internalization of filial piety, while the internalized norms provide an emotional foundation for intergenerational interactions.

## Theoretical contributions and practical implications

This study engages in theoretical dialogue with Levy's Stereotype Embodiment Theory (SET) through an analysis of the formation mechanism of aging imagination among young people. Beyond empirically validating the core hypotheses of SET, this research extends the theory to the Chinese context to examine intergenerational interactions issues. Building on SET's emphasis on individuals' passive internalization of social norms, this study further highlights the influences of relational interactions and cultural construction on the internalization of such norms.

First, this study supports SET's propositions regarding the early internalization of age beliefs and their psychological embodiment through robust empirical data. Levy argues that societal stereotypes about aging begin to be internalized early in life and, as self-relevant schemas, exert profound influences on individuals' perceptions of old age, behaviors, and health outcomes through psychological, physiological, and behavioral pathways [2]. Framing young adulthood as a critical stage for examining the formation of aging-related cognition, this research focuses on young people's aging imagination, which is a concrete manifestation of the early-life internalization of age beliefs as posited in Levy's theory [4]. Empirical findings reveal that grandparent-grandchild communication significantly predicts more positive aging imagination among young adults, essentially identifying an early source of positive age beliefs. Beyond corroborating the notion that the internalization of aging-related cognition commences in early life, this discovery elaborates on the emergence of the embodied effects central to SET from a life-course perspective, providing evidence for the mediating mechanisms through which age beliefs form and subsequently shape future outcomes.

This study also revises SET's emphasis on passive, unidirectional internalization by revealing the active, dynamically constructed nature of age stereotypes within the context of close relationships. SET places greater focus on the unidirectional influence of macro-level, anonymous social narratives—such as portrayals of older adults in media—on individuals' stereotypes [4]. By centering intergenerational interactions within Chinese families, this research challenges this perspective. Findings indicate that intergenerational communication does not directly or mechanically transmit images of "old"; instead, it operates through the critical emotional mediating variable of intergenerational relationship quality. High-quality close relationships foster young adults' willingness to engage in positive dialogue with their grandparents and cultivate empathy, thereby linking "aging" to concrete, vivid individual life experiences and the transmission of wisdom, rather than abstract negative labels. In this way, the internalization of stereotypes is no longer a purely social indoctrination process but a relationally driven, emotionally negotiated one. Close relationships here function as a buffer against, and even a transformer of, negative societal stereotypes, enabling the internalization of positive age beliefs.

This study further revises the cultural applicability of SET by contextualizing it within China's filial piety culture. Both the theoretical content and validation materials of SET originate from Western societies, with limited elaboration on specific Eastern cultural contexts [2]. The present research demonstrates that the internalization of stereotypes is deeply intertwined with cultural norms. In individualism-oriented Western cultures, aging imagination may be more strongly shaped by social welfare systems [73] and media representations [74]. In contrast, within China's collectivism-rooted cultural context, familialism serves as the foundational framework for understanding Chinese society [75]. Filial piety in Chinese culture is not merely an external behavioral norm but a deeply ingrained cultural cognitive schema encompassing intergenerational responsibilities, the value of respecting elders, and the meaning of life [46].This study finds that young adults internalize positive perceptions of aging through intergenerational interactions, with filial piety beliefs functioning as a mediating mechanism. This finding directly challenges the negative narratives

prevalent in Western-centric research that equate "aging" with "frailty" and "burden" [16]. By offering a culture-specific perspective on SET, it suggests that the theory may manifest in distinct forms and operate through different mechanisms across cultural contexts. Furthermore, the serial mediating effect of intergenerational relationships and filial piety beliefs indicates that close intergenerational bonds lay the groundwork for the voluntary, emotional internalization of filial norms. In turn, internalized filial values guide and elevate intergenerational interactions, reinforcing emotional connections. This dynamic cyclical mechanism resonates with both the emphasis on emotional interdependence in Family Systems Theory [45] and the focus on normative integration in Intergenerational Solidarity Theory [60]. It thus highlights that the development of positive age beliefs is the product of the synergistic interplay between affective solidarity and normative solidarity in micro-level family practices.

In summary, this study validates and extends Stereotype Embodiment Theory (SET). It shifts the research focus from macro-level societal contexts to micro-level family dynamics, from passive reception to active construction, and from universal psychological mechanisms to culture-specific pathways. Consequently, this paper offers a novel framework for understanding the developmental mechanisms underlying age-related cognition.

This study further offers implications for aging education in Chinese society. As an emotional communication strategy for intergenerational interactions, intergenerational learning programs have been implemented in numerous countries [76]. Defined as a reciprocal and integrative learning model across age groups, intergenerational learning encourages individuals of different ages to work and learn together, facilitating the exchange of ideas, emotions, experiences, and professional skills. Through this process, participants gain skills, knowledge, and values [77]. Currently, intergenerational learning is also emerging quietly in China. When researching Digital backfeeding, Chinese scholars have found that intergenerational learning centered on Digital backfeeding within Chinese families constitutes a two-way interactive process. The guidance provided by the younger generation to their elders reflects mutual influence between two or even three generations, rooted in emotions, intergenerational relationships, and power structures. Digital backfeeding fosters processual intimacy across generations, which further enhances mutual understanding, emotional alignment, and conceptual consistency among different age groups. [17]. Research from Taiwan also indicates that older adults' life narratives can help reduce emotional distance between generations [3]. To promote intergenerational learning in China, we propose developing targeted programs themed around digital support and oral life histories, and integrating these into school education. Such initiatives would create more natural, equal, and harmonious opportunities for interactions between older and younger generations. As the primary context for aging education, families can promote an intergenerational co-learning model based on voluntariness and equality. While encouraging young adults to assist older generations in learning how to use smart devices, families should also encourage youth to actively listen to elders share life experiences, traditional crafts, or practical wisdom, fostering a two-way flow of Digital backfeeding and experiential inheritance. Additionally, traditional family rituals can be transformed into aging education carriers: during ceremonies such as Spring Festival reunions and birthday celebrations, sessions for life narrative sharing can be added, allowing elders to recount their life stories and converting ritualistic communication into in-depth exchanges of emotional resonance and life cognition. Families can also organize intergenerational agreement activities around family anniversaries, where grandparents and grandchildren jointly define directions for future elderly life planning. Schools should integrate aging education into the national education system by offering life education courses. Through picture book reading, role-playing, and similar activities, students can develop an understanding of the integrity of the life course. Schools can also encourage the organization of "Grandparents'day" events, engaging multiple generations in interactive activities to cultivate mutual respect and empathy across age groups.

## Conclusion and limitations

This study adopts a perspective that transitions from others to the self, systematically examining the pathways through which grandparent-grandchild communication influences young adults' aging imagination within the context of Chinese families. The following conclusions were drawn:

Both daily communication and ritualistic communication were found to significantly and positively influence young adults' aging imagination, with daily communication exerting a stronger effect. Intergenerational relations were shown to mediate the

effects of both communication types on aging imagination, indicating that the quality of intergenerational relations serves as a crucial bridge for attitude transformation among young adults. The mediating effect of filial piety confirmed the cultural-psychological impact within intergenerational interactions, although the strength of this effect varied depending on the communication form. Furthermore, intergenerational relations and filial piety jointly functioned as sequential mediators between intergenerational communication and young adults' aging imagination, revealing a synergistic mechanism between relationship quality and cultural values. These findings provide a theoretical basis for aging education in China.

This study expands the boundaries of intergenerational interactions research by shifting the focus from the elderly to young adults and extending the analysis from individual-level interactions to family systems and cultural transmission. Additionally, the findings offer new perspectives for promoting positive aging in China, suggesting that optimizing intergenerational communication strategies can alleviate young adults' negative perceptions of aging and provide a more constructive approach to understanding personal and societal aging.

However, certain limitations should be acknowledged. Although this study offers insights into how intergenerational interactions influence young adults' aging imagination, these perceptions are part of a dynamic process. The inability to collect longitudinal data on these changing perceptions presents a common limitation in related research [15]. Therefore, ongoing follow-up with some participants is planned to provide deeper insights in future studies. Meanwhile, this study did not measure variables such as family structure, residential distance, and digital media communication. Future research should fully consider the impacts of these factors on the model to gain a more comprehensive understanding of the relationship between grandparent-grandchild family communication and young adults' aging imagination.

## Supporting information

**S1 Table. The chain mediation of interpersonal relations and filial piety among grandchildren and youths (Model 1 without covariates).**
(DOCX)

**S2 Table. The chain mediation of interpersonal relations and filial piety among grandchildren and youths (Model 2 without covariates).**
(DOCX)

## Author contributions

**Conceptualization:** Xiaojuan Hu.

**Data curation:** Xiaojuan Hu.

**Formal analysis:** Xiaojuan Hu.

**Methodology:** Fen Xie.

**Project administration:** Xiaojuan Hu.

**Software:** Xiaojuan Hu.

**Supervision:** Xiaojuan Hu, Fen Xie.

**Writing – review & editing:** Fen Xie.

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
