## [Decision Letter · Decision Letter 0]

9 Dec 2025

Dear Dr. Xie,

Thank you for submitting your manuscript to PLOS ONE. After careful consideration, we feel that it has merit but does not fully meet PLOS ONE’s publication criteria as it currently stands. Therefore, we invite you to submit a revised version of the manuscript that addresses the points raised during the review process.

We look forward to receiving your revised manuscript.

Kind regards,

Cheong Yu Stephen Chan

Academic Editor

PLOS One

Journal Requirements:

“This research was funded by Guangxi Philosophy and Social Science Foundation Project (Nos:23FSH028)”

3. Please note that your Data Availability Statement is currently missing the repository name. If your manuscript is accepted for publication, you will be asked to provide these details on a very short timeline. We therefore suggest that you provide this information now, though we will not hold up the peer review process if you are unable.

4. Please ensure that you refer to Figures 1 to 4 in your text as, if accepted, production will need this reference to link the reader to the figure.

5. We note you have included a table to which you do not refer in the text of your manuscript. Please ensure that you refer to Tables 1 and 2 in your text; if accepted, production will need this reference to link the reader to the Table.

**Additional Editor Comments:**

I endorsed two reviewers' comments. The manuscript needs clearer conceptual framing of “aging imagination,” with a working definition, contrasts to related constructs, and a theoretical model. Methodologically, the authors should separate filial piety’s reciprocal and authoritarian dimensions, provide full item wordings and stronger validity evidence. Results tables require correction of confidence intervals and coefficient reporting. Substantively, the discussion should avoid over-interpretation. Finally, the paper would benefit from practical, evidence-based recommendations for interventions.

Reviewers' comments:

Reviewer's Responses to Questions

**Comments to the Author**

1. Is the manuscript technically sound, and do the data support the conclusions?

Reviewer #1: Yes

Reviewer #2: Partly

2. Has the statistical analysis been performed appropriately and rigorously?

Reviewer #1: Yes

Reviewer #2: No

3. Have the authors made all data underlying the findings in their manuscript fully available?

Reviewer #1: Yes

Reviewer #2: Yes

4. Is the manuscript presented in an intelligible fashion and written in standard English?

Reviewer #1: Yes

Reviewer #2: No

Reviewer #1: 1. The abstract effectively summarizes the study, but it can be enhanced by simplifying some of the language and focusing on the core findings more directly. Additionally, providing a bit more context on the cultural implications of filial piety could help readers unfamiliar with the topic understand the broader significance.

2. The research questions and hypotheses are central to the structure of this study, but there seems to be a mismatch between them that could confuse readers or undermine the clarity of the study's design. Specifically:

Research Question 1 (Q1) focuses on routine (daily) grandparent-grandchild communication and its influence on young people's perceptions of aging.

Research Question 2 (Q2) looks at ritualistic communication and its impact on the same outcome.

However, the hypotheses (H1 and H2) do not fully align with these research questions. Both hypotheses center on intergenerational relations as a mediator, but they do not clearly differentiate between how routine and ritualistic communication might work differently through these relations.

3. The discussion effectively summarizes the study's findings and their contributions to the field, but it would benefit from a more critical and in-depth analysis of the results. Specifically, the discussion could explore the following areas more deeply:

3.1 Theoretical Implications: While the study connects its findings with existing theories like stereotype internalization theory, it would be valuable to expand on how the findings either challenge or extend these theories. For example, how do the specific mechanisms (daily vs. ritualistic communication) align with or contradict existing models of intergenerational communication and aging? Are there any contradictions or nuances that haven't been addressed yet?

3.2 Cultural Context: The study is grounded in the Chinese family context, which is a significant contribution. However, a more critical exploration of filial piety in relation to aging imagination would enrich the discussion. Given that filial piety is a cultural value that varies significantly across different societies, how might these findings differ in other cultural contexts? Could there be alternative interpretations or dimensions of filial piety that the study has not fully addressed?

4. The study discusses a phenomenon in Chinese society, which is valuable. However, it would be helpful to more explicitly explain why this specific focus on Chinese society is important and how the findings contribute to the existing body of literature. For example, what unique insights can this study offer in the context of Chinese cultural values, and how does it extend or challenge existing research on intergenerational communication and aging perceptions? A clearer articulation of the study's contribution will strengthen the significance of the findings and their relevance to broader discussions in the field.

Reviewer #2: The manuscript examines how grandparent–grandchild daily and ritualistic communication relate to Chinese youths’ “aging imagination,” mediated by intergenerational relations and filial piety. This is timely and relevant to gerontology, family sociology, and communication studies. However, this manuscript need to strengthen construct clarity, measurement validity, analytic rigor (controls, robustness), and reporting transparency. Reframe causal language, fix table inconsistencies, and improve data availability.

Major comments with detailed suggestions

1. Conceptual clarity and theoretical framing. Regarding the definition and scope of “aging imagination,” the construct appears to mix self-oriented imagined aging (e.g., planning, acceptance) with broader attitudes toward aging. The current description does not distinguish it from adjacent constructs (e.g., ageism, aging anxiety, future time perspective). I suggest providing a working definition, explicitly contrasting it with related constructs, and including a brief theoretical model figure that positions aging imagination among them.

2. The authors argue that daily communication has stronger effects than ritualistic communication, primarily on frequency grounds. They should expand the mechanism by drawing on social learning, intimacy, and role reciprocity; explicitly hypothesize moderators that shape relative effects (e.g., co-residence, geographic proximity, migration status, digital media substitution for rituals, quality of family climate); and add a paragraph explaining how ritualistic contact can be potent in value-activation contexts (e.g., during key rites), potentially boosting the normative (authoritarian) component of filial piety.

3. The authors discuss filial piety as a dual construct (reciprocal vs. authoritarian) but analyze it as a single composite. They should separate and report the two subdimensions.

4. The manuscript does not include full item wordings for all constructs, and the reported psychometric evidence is limited. Please provide full item lists and additional validity evidence.

5. Harman’s single-factor test is inadequate on its own. Add a CFA-based test comparing a one-factor model with the proposed multi-factor model and report the change in fit indices.

6. No covariates are included, so potential confounds remain. Report models with and without controls to demonstrate robustness of the mediation effects. At a minimum, include plausible confounders and family-structure controls. Research design and inference should reflect these additions.

7. There is an inconsistency about recruiting “experts and practitioners” versus “young people,” and the exclusion criteria are vague. Please clarify the sampling frame and operationalize exclusions.

8. In Tables 5 and 6, the direct-effect confidence intervals are inconsistent with the point estimates, and it is unclear whether coefficients are standardized or unstandardized. Please correct and clarify.

9. Avoid over-interpretation and offer more nuanced discussion. Recast the discussion to emphasize associations and plausible mechanisms. Integrate literature on digital intergenerational contact (e.g., whether WeChat-based contact substitutes for or complements face-to-face interactions), and note potential negative dynamics (e.g., ritual obligations perceived as pressure may undermine positive aging imagination for some youth).

10. Provide practical recommendations with guardrails. For example, in educational interventions, propose short, evidence-based modules that promote reflective dialogue with grandparents (e.g., guided oral history, co-learning technology tasks), rather than merely advocating “more contact.”

11. Add the IRB approval number/code if available.

12. Regarding writing and formatting, clean up spacing and punctuation throughout (e.g., change “relations ,” and “youths’constructive” to “relations,” and “youths’ constructive”).

.

Reviewer #1: No

Reviewer #2: No

---

## [Author Response · Author response to Decision Letter 1]

2 Feb 2026

Dear editor and Reviewers,

Thank you very much for the opportunity to revise our manuscript entitled “The Impact of Grandparent–Grandchild interactions on the Imagination of Aging among Chinese Youth Groups: The Chain Mediating Role of Intergenerational Relations and Filial Piety Concept” (Manuscript No. PONE-D-25-46690). We sincerely appreciate the time and effort you and the reviewers have devoted to evaluating our work. We have carefully considered all comments from editor/Reviewer #1 and Reviewer #2, and have undertaken substantial revisions to strengthen the manuscript conceptually, methodologically, and substantively.Below, we provide a detailed response to each of their comments. All revisions have been highlighted in blue/red/purple in the revised manuscript.

—Response to editor

Comment 1. Please ensure that your manuscript meets PLOS ONE's style requirements, including those for file naming. The PLOS ONE style templates can be found at https://journals.plos.org/plosone/s/file?id=wjVg/PLOSOne_formatting_sample_main_body.pdf and https://journals.plos.org/plosone/s/file?id=ba62/PLOSOne_formatting_sample_title_authors_affiliations.pdf

Response:

We have revised the format in accordance with the template requirements of PLOS ONE.

Comment2. Thank you for stating the following financial disclosure: “This research was funded by Guangxi Philosophy and Social Science Foundation Project (Nos:23FSH028)”Please state what role the funders took in the study.  If the funders had no role, please state: "The funders had no role in study design, data collection and analysis, decision to publish, or preparation of the manuscript." If this statement is not correct you must amend it as needed. Please include this amended Role of Funder statement in your cover letter; we will change the online submission form on your behalf.

Response：

This research was funded by the Guangxi Philosophy and Social Science Foundation (No. 23FSH028) and the Guangxi Normal University Project (No. 2024PY004). The corresponding author of this paper is the principal investigator of the project, and the first author was the primary contributor to its implementation. The survey was conducted with support from these projects, and the article processing charges (APCs) are also covered by them. Thus, the study received full financial support from both projects.

We have already provided a detailed explanation in the cover letter.

Comment3. Please note that your Data Availability Statement is currently missing the repository name. If your manuscript is accepted for publication, you will be asked to provide these details on a very short timeline. We therefore suggest that you provide this information now, though we will not hold up the peer review process if you are unable.

Response：

We have now provided the name and link to the database.The data that support the findings of this study are available on request from OPEN ICPSR(https://www.openicpsr.org/openicpsr/workspace?path=openICPSR(Project ID:openicpsr-222102)).

Comment4. Please ensure that you refer to Figures 1 to 4 in your text as, if accepted, production will need this reference to link the reader to the figure.

Response：

We have marked the figures and tables at the relevant positions in the text and cross-checked by two authors

Comment5. We note you have included a table to which you do not refer in the text of your manuscript. Please ensure that you refer to Tables 1 and 2 in your text; if accepted, production will need this reference to link the reader to the Table.

Response：

We have labeled Table 1 and Table 2 at the relevant points in the text.

Comment6. Please include captions for your Supporting Information files at the end of your manuscript, and update any in-text citations to match accordingly. Please see our Supporting Information guidelines for more information: http://journals.plos.org/plosone/s/supporting-information.

Response：

We have added a title for supporting information at the end of the article，Please refer to the relevant sections of the article for specific details.

Comment7. If the reviewer comments include a recommendation to cite specific previously published works, please review and evaluate these publications to determine whether they are relevant and should be cited. There is no requirement to cite these works unless the editor has indicated otherwise.

Response：

We have cited the literature based on the actual needs of the article, and there are no redundant or irrelevant citations.

—Below are the responses to Reviewer 1 and Reviewer 2, with the corresponding revisions in the text highlighted in blue or red.

Response to Reviewer 1

Comment 1:The abstract effectively summarizes the study, but it can be enhanced by simplifying some of the language and focusing on the core findings more directly. Additionally, providing a bit more context on the cultural implications of filial piety could help readers unfamiliar with the topic understand the broader significance.

Response：

We fully agree that simplifying language, centering core findings, and adding cultural context for filial piety would enhance clarity and accessibility for all readers—particularly those unfamiliar with Chinese cultural norms.

We have revised the abstract to address the key suggestions: simplifying language, centering core findings more directly, and enhancing cultural context for filial piety to improve accessibility for readers unfamiliar with Chinese cultural norms.

The revised abstract now foregrounds the cultural framework of Confucian filial piety, explicitly defining it as a core norm centered on respecting elders and intergenerational responsibility, while tightly linking it to China’s population aging context.

We have refined the academic expressions, elaborated on the effects of daily communication and ritualized interaction—the core finding of the study—and clarified the pivotal roles of intergenerational relations and filial piety in both the dual mediating effect and the serial mediating mechanism. The revised version maintains brevity while emphasizing practical implications, ensuring readers grasp the study’s core contributions at a glance.

The following is the final revised abstract, which has been incorporated into the manuscript and appears on【Lines 18 to 29 of the text】.

“Centered on respecting elders and intergenerational responsibility, Confucian filial piety culture serves as a core cultural norm in Chinese society. This study integrates this cultural framework with the context of China’s rapid population aging to explore how grandparent-grandchild interactions within Chinese families shape young adults’ aging imagination. Based on a survey of 778 Chinese youth aged 18-35, the findings show that both daily and ritualistic interactions positively predict young adults’ aging imagination, and daily interactions has a stronger predictive effect. Intergenerational relations and filial piety not only serve as independent mediators but also constitute a chain mediating pathway, indicating that emotional bonds and cultural values synergistically foster young people’s positive aging imagination. This study suggests that efforts to strengthen intergenerational interactions and integrate culturally rooted aging education are critical for promoting positive aging attitudes among youth.”

Comment 2. The research questions and hypotheses are central to the structure of this study, but there seems to be a mismatch between them that could confuse readers or undermine the clarity of the study's design. Specifically:

Research Question 1 (Q1) focuses on routine (daily) grandparent-grandchild communication and its influence on young people's perceptions of aging.

Research Question 2 (Q2) looks at ritualistic communication and its impact on the same outcome.

However, the hypotheses (H1 and H2) do not fully align with these research questions. Both hypotheses center on intergenerational relations as a mediator, but they do not clearly differentiate between how routine and ritualistic communication might work differently through these relations.

Response：

We fully agree your opinion. To address this problem, we have revised the study’s conceptual framework to unify all research aims under hypotheses, ensuring clarity, consistency, and rigor in the study design.

Specific Revisions:

Eliminated standalone research questions (Q1-Q2): We replaced the question-based framing with explicit hypotheses to directly link each communication type (daily/ritualistic) to the outcome (aging imagination) and clarify their unique pathways.

Restructured and renumbered hypotheses for alignment:

Original Q1 (daily interactions→aging imagination) is now Hypothesis 1: Daily interactionss between grandparents and grandchildren significantly influences the aging imagination of young people. 【See line 173 of the text】.

Original Q2 (ritualistic interactionss→aging imagination) is now Hypothesis 2:Ritualistic interactions between grandparents and grandchildren significantly influences the aging imagination of young people. 【See line 175 of the text】.

Original H1-H6 are renumbered as Hypotheses 3-8, with each explicitly specifying the communication type they address to eliminate ambiguity:

H3:Intergenerational relations mediate the association between daily grandparent-grandchild interactions and young peoples’aging imagination of aging. 【See line 205 of the text】.

H4: Intergenerational relations mediate the association between ritualistic grandparent-grandchild interactions and young adults’ aging imagination.【See line 207 of the text】.

H5: Filial piety mediates the association between daily grandparent-grandchild interactions and young adults’ aging imagination.【See line 246 of the text】.

H6: Filial piety mediates the association between ritualistic grandparent-grandchild interactions and young adults’ aging imagination.【See line 248 of the text】.

H7: Intergenerational relations and filial piety sequentially mediate the association between daily grandparent-grandchild interactions and young adults’ aging imagination.【See line 285 of the text】.

H8: Intergenerational relations and filial piety sequentially mediate the association between ritualistic grandparent-grandchild interactions and young adults’ aging imagination.【See line 288 of the text】.

Comment 3. The discussion effectively summarizes the study's findings and their contributions to the field, but it would benefit from a more critical and in-depth analysis of the results. Specifically, the discussion could explore the following areas more deeply:

3.1 Theoretical Implications: While the study connects its findings with existing theories like stereotype internalization theory, it would be valuable to expand on how the findings either challenge or extend these theories. For example, how do the specific mechanisms (daily vs. ritualistic communication) align with or contradict existing models of intergenerational communication and aging? Are there any contradictions or nuances that haven't been addressed yet?

3.2 Cultural Context: The study is grounded in the Chinese family context, which is a significant contribution. However, a more critical exploration of filial piety in relation to aging imagination would enrich the discussion. Given that filial piety is a cultural value that varies significantly across different societies, how might these findings differ in other cultural contexts? Could there be alternative interpretations or dimensions of filial piety that the study has not fully addressed?

Response:

Thank you for your profound insights on the discussion section of the manuscript. We fully agree that these comments will significantly enhance the quality of the paper. Below is a detailed explanation of the targeted revisions we have made in response to these points.

We engaged in theoretical dialogue with Stereotype Embodiment Theory (SET) by analyzing how our findings validate, extend, and revise the theory:

(1) Validating the Core Proposition of Stereotype Embodiment Theory (SET)

We confirmed the core tenet of SET regarding the early internalization of age-related beliefs (Levy, 2009). Our findings reveal that grandparent-grandchild communication, as an early source of age-related experiences, influences young adults’ aging imagination—an explicit manifestation of internalized age schemas (Diel & Wahl, 2024). This provides empirical support for SET’s claim that "age stereotypes become embodied through life course processes."

(2) Extension: From Passive Internalization to Active Relational Construction

SET primarily conceptualizes stereotype internalization as a unidirectional process driven by macro-social narratives (e.g., media portrayals; Wu, 2021). Our results challenge this perspective by demonstrating that grandparent-grandchild communication operates through the mediating role of intergenerational relationship quality. High-quality emotional bonds encourage young adults to proactively engage with and empathize with their grandparents, transforming abstract social stereotypes into concrete, positive aging imaginations (e.g., the transmission of wisdom). This finding extends SET by illustrating, at the family level, how positive intergenerational interactions and relationships can mitigate negative aging perceptions.

(3) Cultural Revision of Stereotype Embodiment Theory (SET)

This study also revises the cultural applicability of SET by contextualizing it within Chinese filial piety culture. SET’s theoretical framework and empirical validation are rooted in Western societies, with limited elaboration on specific Eastern cultural contexts. Our research demonstrates that the process of stereotype internalization is intertwined with cultural norms. In individualistic Western cultures, aging imagination may be more heavily influenced by social welfare systems and media discourse. In the Chinese context, familism serves as the foundation for understanding Chinese society (Yu, 2023), and filial piety (xiao) is not merely an external behavioral norm but a deeply ingrained cultural cognitive schema encompassing intergenerational responsibility, respect for elders, and the meaning of life (Yeh & Bedford, 2003). Our findings show that, through the mediating role of filial piety, young adults internalize positive perceptions of aging during intergenerational interactions, offering a culture-specific perspective to SET.Furthermore, the chain mediating effect of intergenerational relations and filial piety confirms that close emotional bonds lay the groundwork for the voluntary, affect-driven internalization of filial norms. In turn, internalized filial values guide and elevate intergenerational interactions, strengthening emotional connections. This finding aligns with Family Systems Theory’s emphasis on emotional interdependence and Intergenerational Solidarity Theory’s focus on normative integration, indicating that the formation of positive age beliefs is the result of the synergistic interplay between affectual solidarity and normative solidarity in micro-level family practices.

In summary, this study validates and extends Stereotype Embodiment Theory by shifting the research focus from macro-society to micro-family contexts, from passive reception to active construction, and from universal psychological mechanisms to culture-specific pathways. It offers a novel framework for understanding the developmental mechanisms of age cognition.

【The specific content can be found on Lines 701 to 767 of the text】

“This study engages in theoretical dialogue with Levy’s Stereotype Embodiment Theory (SET) through an analysis of the formation mechanism of aging imagination among young people.......In summary, this study validates and extends Stereotype Embodiment Theory (SET). It shifts the research focus from macro-level societal contexts to micro-level family dynamics, from passive reception to active construction, and from universal psychological mechanisms to culture-specific pathways. Consequently, this paper offers a novel framework for understanding the developmental mechanisms underlying age-related cognition.”

Comment 4. The study discusses a phenomenon in Chinese society, which is valuable. However, it would be helpful to more e

---

## [Decision Letter · Decision Letter 1]

11 Mar 2026

The impact of grandparent-grandchild interactions on the imagination of aging among Chinese youth groups: The chain mediating role of intergenerational relations and filial piety concept

PONE-D-25-46690R1

Dear Dr. Xie,

We’re pleased to inform you that your manuscript has been judged scientifically suitable for publication and will be formally accepted for publication once it meets all outstanding technical requirements.

Kind regards,

Cheong Yu Stephen Chan, Ph.D.

Academic Editor

PLOS One

Additional Editor Comments (optional):

Reviewers' comments:

Reviewer's Responses to Questions

**Comments to the Author**

Reviewer #1: All comments have been addressed

Reviewer #2: All comments have been addressed

2. Is the manuscript technically sound, and do the data support the conclusions?

Reviewer #1: Yes

Reviewer #2: Yes

3. Has the statistical analysis been performed appropriately and rigorously?

Reviewer #1: Yes

Reviewer #2: Yes

4. Have the authors made all data underlying the findings in their manuscript fully available?

Reviewer #1: Yes

Reviewer #2: Yes

5. Is the manuscript presented in an intelligible fashion and written in standard English?

Reviewer #1: Yes

Reviewer #2: Yes

Reviewer #1: (No Response)

Reviewer #2: I appreciate the hard work of the authors. All of the comments have been addressed in this version. I have no further comments on this manuscript.

.

Reviewer #1: No

Reviewer #2: **Yes:** Cheung Vanessa Hoi MeiCheung Vanessa Hoi MeiCheung Vanessa Hoi MeiCheung Vanessa Hoi Mei

---

## [Editor Report · Acceptance letter]

PONE-D-25-46690R1

PLOS One

Dear Dr. Xie,

I'm pleased to inform you that your manuscript has been deemed suitable for publication in PLOS One. Congratulations! Your manuscript is now being handed over to our production team.

Kind regards,

on behalf of

Dr. Cheong Yu Stephen Chan

Academic Editor

PLOS One